# Applying the Fit-for-Purpose Land Administration Concept to South Africa †

**Christopher Williams-Wynn**

Surveyor-General, Eastern Cape, Department of Agriculture, Land Reform and Rural Development, East London, Pretoria 0001, South Africa; Christopher.WilliamsWynn@dalrrd.gov.za; Tel.: +27-82-577-5574

† Initial concepts of this paper were submitted to the Fédération Internationale des Géomètres (FIG) Working Week, Amsterdam, The Netherlands, 10–14 May 2020. Draft proceedings of the cancelled conference were published on 14 April 2020. Available online: https://www.fig.net/resources/proceedings/fig_proceedings/fig2020/papers/ts03h/TS03H_williams-wynn_10363.pdf.

**Abstract:** What potential will the fit-for-purpose land administration concept have of working in the Republic of South Africa? This question is asked against the existence of a high-quality cadastre covering most of the South African landmass. However, a large proportion of the people living in South Africa live outside of this secure land tenure system. Many citizens and immigrants reside on communal land, in informal settlements, in resettled communities, in off-register housing schemes, and as farm dwellers, labour tenants and other occupants of commercial farms. Reasonable estimates suggest that there are more than 5 million land occupations that exist outside the formal land tenure system and hence outside the formal land administration system. This paper looks at the current bifurcated system and considers how the application of the fit-for-purpose land administration system can expand the existing cadastral system and provide security of tenure that is beneficial and acceptable to all. It demonstrates that, not only could it work, but it is also considered to be necessary. This paper uses South Africa as a case study to demonstrate how adjustments to institutional, legal and spatial frameworks will develop a fully inclusive, sufficiently accurate land administration system that fits the purpose for which it is envisioned. These country-specific proposals may well be of international interest to assist with the formulation of fit-for-purpose land administration systems being developed in other countries.

**Keywords:** fit-for-purpose land administration; spatial; legal; and institutional frameworks; land tenure security; pro-poor land recordation; land governance reform; cost effectiveness; innovative technology

## 1. Introduction

Land administration is defined in the Land Administration Domain Model as "the process of determining, recording and disseminating information about the relationship between people and land" [1]. The Framework for Effective Land Administration released by the United Nations Committee of Experts on Global Geospatial Information Management notes that "all people have the right to an adequate standard of living, regardless of whether underlying people-to-land relationships are formal, informal, statutory, customary, legal, legitimate, or otherwise in nature" [2], (p. 7). The Voluntary Guidelines on the Responsible Governance of Tenure of Land, Fisheries and Forests in the context of National Food Security (VGGT) notes that "access to land, fisheries and forests is defined and regulated by societies through systems of tenure. These tenure systems determine who can use which resources, for how long, and under what conditions" [3], (p. iv). Therefore, one of the guiding principles of responsible tenure governance (a key element of land administration) is to "promote and facilitate the enjoyment of legitimate tenure rights" [3], (p. 3), and equitable access to land, fisheries and forests. This should be applicable to all

forms of land tenure, whether it be public, private, communal, indigenous, customary, or informal [3], (p. 7).

The International Land Measurement Standard describes "Land Tenure" as "the rules and arrangements connected with owning specified interests in the land. This can be defined as the relationship, whether legally or customarily defined, among people, as individuals or groups, with respect to land and associated natural resources (water, trees, minerals, wildlife, etc.). Rules of tenure define how property rights in land are to be allocated within societies. Land tenure systems determine who can use what resources for how long, and under what conditions" [4] (p. 24).

Enemark and McLaren highlight that, while there exists a wealth of literature that "emphasises the need for security of tenure and elaborates on its benefits, including the opportunities of significantly contributing to poverty reduction and sustainable development, the conventional approaches to land do not make this a reality" [5], (p. 3). Conventional land administration systems require high accuracy standards for identification, mapping and recordation of land rights. They are generally expensive and operate within a judicially oriented legal framework. As an alternative approach to conventional land administration, the fit-for-purpose land administration concept considers the cultural, social, economic and political context of a country and builds the components of land administration to benefit all people, regardless of their economic or social status [6], (p. 6). In recording land occupation and use, it recommends the use of visible features rather than invisible boundaries based on monumentation [5], (p. 21). It promotes the use of modern (advancing and affordable) technology such as geographic information system mapping technology (GIS), rectified imagery (computer software-generated true-scale aerial photographs) and Global Navigation Satellite System position fixing (GNSS) [6], (pp. 16–17) and [5], (p. 32). It advocates that adjudication, recordation and dispute resolution should be handled through transparent, flexible and simple administrative procedures [5] (p. 27), utilising a human rights approach with all interested and affected parties participating.

The VGGT emphasises that a secure tenure system supports the recognition and respect of all legitimate (formal and social) tenure right holders and their rights, promotes the safeguarding of their rights against threats and infringements and promotes access to justice, thereby minimising tenure disputes, violent conflicts and corruption [3], (p. 3). All people who legitimately occupy land should be provided with a form of secure land tenure that is affordable resulting from highly participatory, quick and efficient methods of recordation, and which can be incrementally improved whenever desired [5], (pp. 3, 5). The fit-for-purpose land administration concept with its three key pillars (institutional, legal and spatial frameworks) supports all these goals by maximising the documenting and recording of people-to-land relationships, thereby facilitating their recognition and inclusion [7], (p. 13).

## 2. Research Problem and Methodology

The publication "Fit-for-purpose Land Administration: Guiding Principles for Country Implementation" sets out certain principles for consideration. Firstly, the pro-poor fit-for-purpose approach "will lead to social inclusion, increased equity and better recognition of human rights" [7], (p. 5). Secondly, land administration functions "include the areas of: land tenure (securing and transferring rights in land and natural resources); land value (valuation and taxation of land and properties); land use (planning and control of the use of land and natural resources); and land development (implementing utilities, infrastructure, construction works, and urban and rural developments)" [7], (p. 9). Thirdly, "there is a consensus that governing the people-to-land relationship is at the heart of the global agenda and that there is an urgent need to build appropriate and basic systems using a flexible and affordable approach to identify the way land is occupied and used by all, whether these land rights are legal or locally legitimate" [7], (p. 13).

The Fédération Internationale des Géomètres (FIG) guide on Fit-for-purpose Land Administration (Publication No. 60) adds that "Fit-for-purpose means that the land ad-

ministration systems—and especially the underlying spatial framework of large-scale mapping—should be designed for the purpose of managing current land issues within a specific country or region" [8] (p. 6). However, comparing the current land administration system of the case study area, i.e., the Republic of South Africa, with the fit-for-purpose land administration concept, reveals two major problem areas that require consideration.

- The existing land administration system of South Africa is constructed on the foundation of the official cadastral records, many of which are old, outdated and prepared before standards existed. Many documented land parcels are therefore inaccurate, especially where the boundaries are based on topographical features that are dynamic in nature. A solution needs to be found to improve existing boundary records through utilising innovative technology and datasets that are readily available.
- Many legitimate land occupations are excluded from the formal land administration system, especially those undocumented rights on communal land, in informal settlements, in resettled communities, in off-register housing schemes and housing of farm dwellers, labour tenants and other occupants of commercial farms. The land administration records are therefore incomplete. A solution needs to be found to bring existing legitimate land occupations into the country's land administration system.

Recognising that South Africa is looking to overcome past racially based inequity of land distribution to achieve socioeconomic stability and inclusive economic growth for all South Africans, the fit-for-purpose land administration concept is being considered to close these gaps. Officials from the Tenure Reform and Spatial Planning components of the South African government have proposed the development of an integrated Land Administration System that:

- Includes a legally secure tenure for those with insecure tenure;
- Promotes socioeconomic stability and growth;
- Develops an efficient land management system that is relevant to all;
- Effects a unitary, non-racial and flexible land tenure system that supports an equitable redistribution of land resources; and
- Links all people to their *indawo*—as it is called in isiZulu (i.e., the land they occupy, use or to which they have rights).

Almost all the landmass of South Africa has been covered by land parcels delineated on diagrams kept in the Offices of the Surveyors-General [9] and registered in deeds effected in the Deeds Registries [10]. These are the two key components of its cadastral system and the foundation of the land administration system. The methodology of this research is, therefore, to analyse the current South African Land Administration System in relation to the institutional, legal and spatial frameworks as constituted by the fit-for-purpose land administration concept [7], (pp. viii and 17). These analyses require an understanding of the history and evolution of the South African land administration system (Section 3) and the resulting current land administration system (Section 4). Analyses and solutions are provided within each of the three frameworks (Sections 5–7), and recommendations on the way forward are provided (Section 8), leading to some conclusions (Section 9). It is posited that the fit-for-purpose land administration concept [7,8] can address the identified problems through careful consideration and standardised application. To implement the South African government's vision, it is argued that the fit-for-purpose land administration concept is highly suited to provide a mechanism to bring all legitimate land occupations, currently excluded, into a unitary land administration system.

### 3. A Brief History of the Land Administration System in the Case Study Area: South Africa

The Republic of South Africa uses a cadastral system introduced during the Dutch occupation of the Cape, based on Roman-Dutch law and brought to the Cape in 1652 by the *Vereenigde Oostindische Compagnie* (VOC) [11], (p. 42). The first land parcels were granted by the Dutch authorities to former employees of the VOC who originated from Europe but

chose to remain in the Cape on conclusion of their term of service. They were known as "free *burghers*" and were given land on condition they assisted in the production of fresh food to supply passing trade ships. The grants were recorded on diagrams (identifying position and extent) and in deeds (text describing the link of the land parcel to the *burgher*). When the British took over the Dutch colony in 1806, all the land within the colony, excluding the existing registered land parcels belonging to the *burghers*, was proclaimed as belonging to the (British) Crown.

Before the establishment of a fully inclusive government in South Africa in 1994, the right to an adequate standard of living resulting from the use and enjoyment of a land parcel within a secure land tenure system excluded most of the indigenous people and enforced separation of race groups. This land administration system was dominated by European (initially Dutch and later British) colonial and ultimately apartheid policies, laws and practices. Fisher and Whittal note that: "Indigenous, native, Asian, ethnically-mixed and freed-slave descendent South Africans have borne the brunt of cruel systems of governance that used land administration as a tool to engineer society along racial lines" [11], (p. 347). Only between 1895 and 1923 were attempts made to extend land rights to a few indigenous peoples of South Africa, in recognition of faithful service. Apart from this brief extension of the land administration system to include people of colour, most occupation by those who did not have European ancestry was restricted to communal land (Figure 1 and Figure S1 [12] in the Supplementary Materials), reserves and dormitory townships.

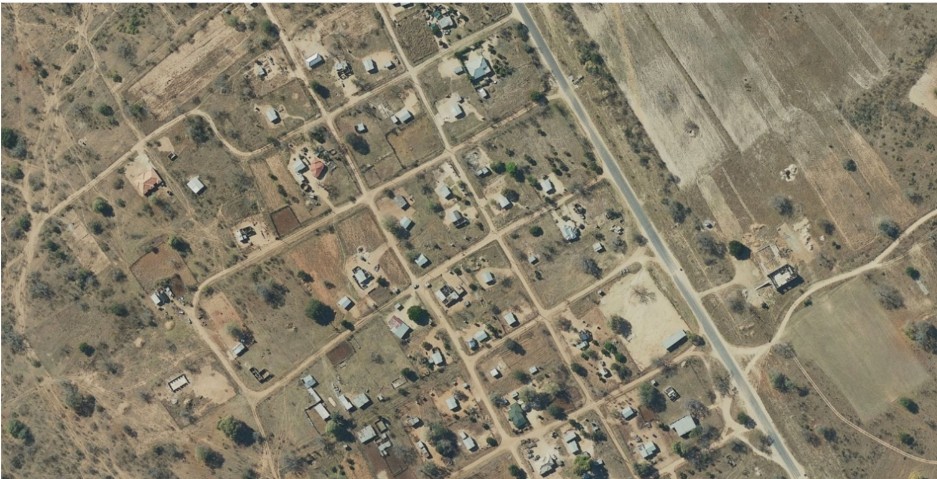

**Figure 1.** Extract from 2018 rectified imagery [12], (Ref: 2531 AC 10), showing part of a rural village known as Buyelani on communal land in the Mpumalanga Province, at scale: 1:5000 (approximate).

Roughly 16 million hectares (or 13% of the South African landmass) was set aside as communal land, an umbrella term for customary land, land under traditional authorities, tribal land and any land that was part of the South African homeland system [13], (p. 13). Even though communities occupy it, communal land comprises land registered as state land, land proclaimed as state land, land assumed to be state land (land that has never been registered), land resumed by the state and "Trust" land. Records of any rights of individual occupation within communal areas are mostly informal and insecure.

Areas of communal land were not self-sustainable and by as early as 1961, they were "overcrowded, and not one of them could feed its own population except for food purchased by wages from outside" [14], (p. 60). As a result, many people migrated to the cities, towns, mines and farms in search of work and wages to feed their families. The migration has not only occurred from communal areas within the South African borders. Migrants from neighbouring countries, in search of a better life, number in the millions; many have relinquished their original nationality and assimilated into local communities to obtain South African citizenship; others have obtained the required work

permits for foreign nationals. Thus, in addition to the insecure tenure pervasive in areas of communal land, insecure tenure is also prevalent in:

- Informal settlements (not only shack dwellers, because informal settlements can include some substantial residences), mostly on the outskirts of urban areas;
- Communities that have been resettled on state-acquired commercial farms as part of the government's redistribution policy where, if the land was transferred to the community, it was transferred to a communal property association;
- Government-initiated housing schemes that were developed in an organised fashion (Figure 2 and Figure S2 [12] in the Supplementary Materials), but where title deeds could not be issued to beneficiaries because of administrative hindrances;
- People that reside on commercial farms as farm workers or labour tenants (i.e., people who may occupy a portion of a farmer's land in exchange for labour); and
- "Ingonyama Trust land", which is an anomaly of communal land that many in government still consider to be state land. Approximately 2.8 million hectares of land is held in trust by the Zulu monarch for the benefit of the Zulu nation. Some leasehold titles have been granted by the Ingonyama Trust Board to residents and business sites.

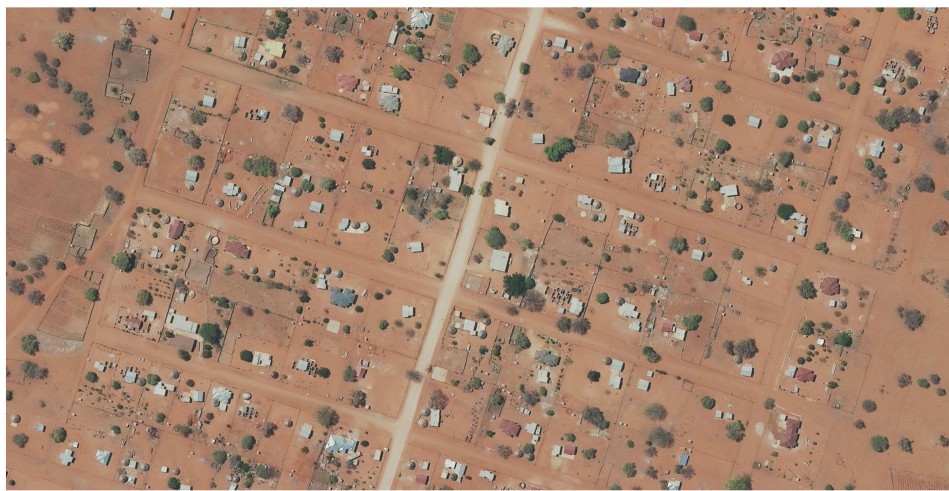

**Figure 2.** Extract from 2018 rectified imagery [12], (Ref: 2230 BC 22), showing part of a rural village known as Tshiungani, a government-initiated settlement scheme in the Limpopo Province, at scale: 1:5000 (approximate).

Figure 3 gives a good indication of population densities, where vast rural areas of the country have occupation densities of 30–300 people per square kilometre, but economically sustainable commercial farms are often larger than three square kilometres (300 hectares). Most of the darker-washed areas to the north and east of the figure is communal land and contains an estimated 2–3 million existing homesteads and other rights. The actual numbers of informal dwellings are unknown, but estimates suggest that similar numbers of dwellings as have been calculated in communal areas now also exist in informal settlements. This means that any proposals to bring the occupants of insecure tenure into a land administration system would possibly need to consider identifying more than 5 million land occupations. The numbers continue to increase.

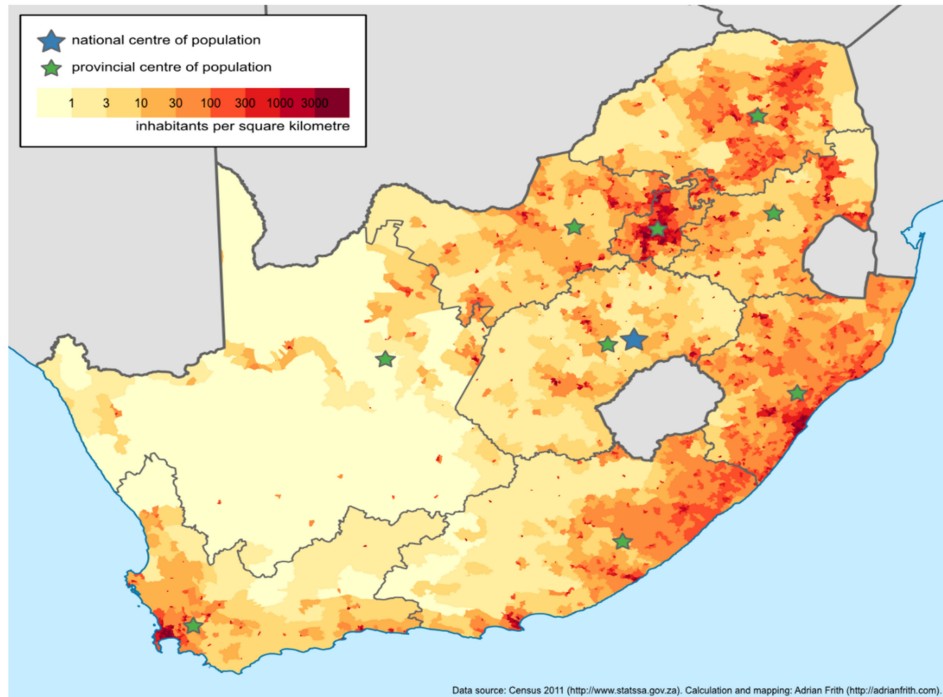

**Figure 3.** Map by Adrian Frith [15], indicating the number of inhabitants per square kilometre in South Africa as recorded in the last census in 2011.

## 4. The Resultant Land Administration System of the Case Study Area: South Africa

The biggest hurdle in South Africa is the multiplicity of different forms of land tenure (formal and less formal) that exist, often overlapping in areas where people-to-land relationships of a community are to be formalised.

- There are rights of the original title holder and successors in title, which were originally granted to people of European descent (known as "whites").
- There are instances of quitrent title, many of which are still recorded in the original Indigenous owner's name (known as "non-white"). Quitrent title was land granted by the Crown to loyal subjects—both European and Indigenous—on condition that the holder would supply the colonial administration a specified annual contribution, either in cash or in farm produce. Although no new quitrent titles were created after 1923 [16], the system was maintained by the state until 1934, when all quitrents belonging to "whites" were converted to freehold title and the remaining quitrent titles held by "non-whites" were, thereafter until recently, ignored.
- Less secure individual rights were granted to indigenous peoples by a Resident Magistrate under the formal "Permission to Occupy" (PTO) system. This system became the jurisdiction of the "*Bantu* Administration" system as in the example in Figure 4. In this example, the arable allotment (No. 50) referred to thereon has a quitrent diagram in the office of the Surveyor-General [16], (Reference S.G. No. 9154/1901) but was issued as a PTO right by an organ of state that no longer exists. It is not registered in the Deeds Registry.
- From 1939, policy succeeding quitrent title and running concurrently with the PTO policy was the "Betterment Scheme" policy, where state officials (usually Agricultural Extension Officers) relocated indigenous people from their smallholdings of arable land (often in extent of between three and seven hectares, with a further right to communal grazing), onto one acre (roughly 2000 square metre) sites in "Villages" (Figure 5) in an attempt to increase the efficiency of agricultural production through state-run cultivation of arable land and consolidation of grazing land.

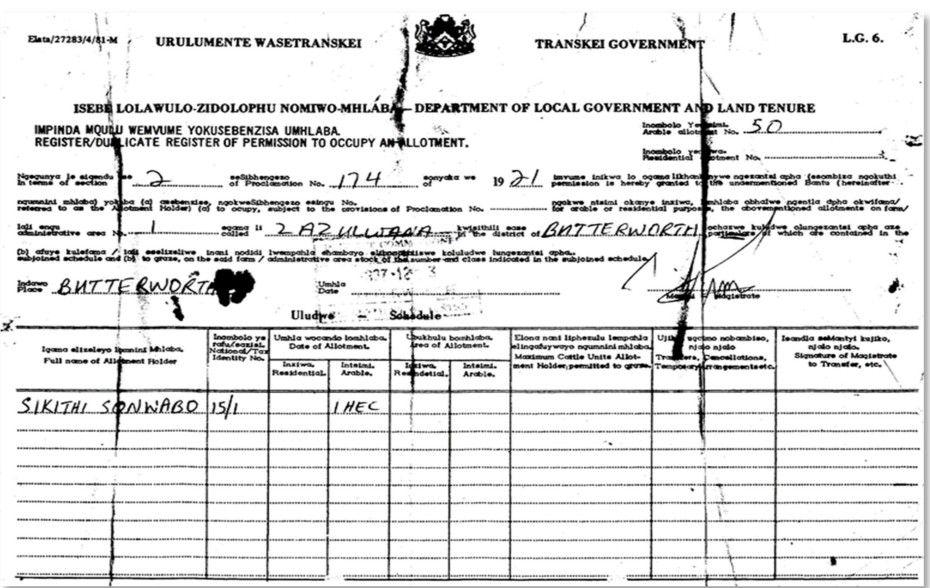

**Figure 4.** A duplicate copy of a Permission to Occupy (PTO) obtained from the owner, Mr Sonwabo Sikithi, on the 26 October 2020. His PTO of 1 hectare is situated in the Administrative area of Zazulwana, District of Butterworth, Eastern Cape Province.

- The government-run PTO system floundered due to lack of maintenance [17], (p. 21), mainly because deaths and succession were seldom reported. There are suggestions that unwillingness to report changes was because "non-white" holders objected to having to continue with their annual contributions, when their "white" counterparts did not, due to the conversion of their rights to freehold. As a result, the African traditional leadership hierarchy (king, chief, headman or council) assumed the responsibility to apportion land to their subjects as they saw fit and were recognised only for as long as the recipient was considered a faithful subject of that traditional authority. These PTOs were usually not recorded in any government-administered system and hence were allocated without reference to existing demarcated rights [17], (pp. 18–19).

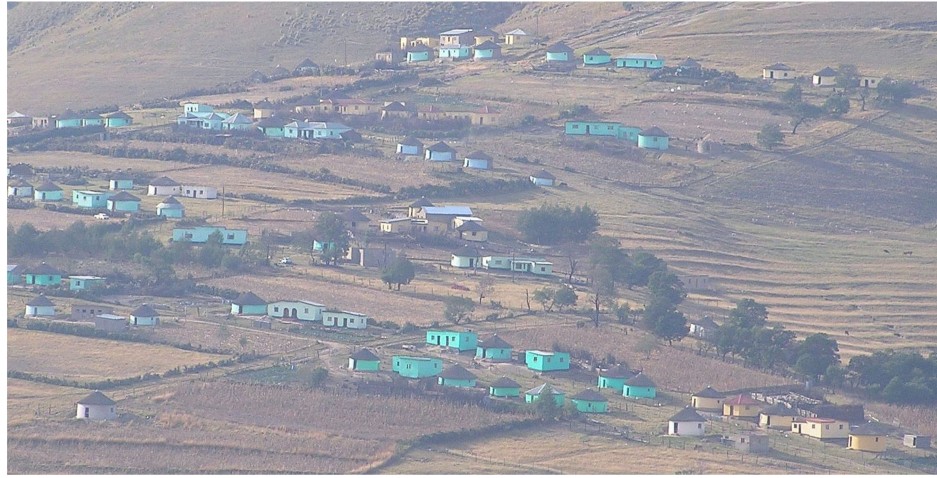

**Figure 5.** Photograph of a "Betterment Scheme" settlement known as Brook's Nek, on communal land in the former Transkei area of the Eastern Cape Province [18].

- Allocations were also determined by civil society and political structures, including "people farmers" and "land grabbers", especially where the traditional leadership was inefficient, or the settlement was outside the communal areas. These land allocations

were often in recognition of membership or allegiance to a specific structure and were often orchestrated as mass invasions of a preidentified land area.

- Informal rights to land would extend to every person currently residing within the area of the community, no matter how they got there; whether by birth, voluntary migration, job seekers, forced resettlement, assimilation or sworn allegiance.

- Those rights may well extend beyond every person residing on the land. Traditional communities recognise the association of all descendants willing to maintain allegiance and therefore as having rights in the area. For example, Bishop Emeritus Andile Mbete detailed to the author that he has a "house" in a suburb of East London, a city in the Eastern Cape Province, but his "home" is near Willowvale, in a communal area in the former Transkei "homeland", from whence he and his ancestors came.

Over the years, all existing deeds (to the exclusion of most other land rights), issued under any administration, have been incorporated into the current deeds registration system, which is now governed by the Deeds Registries Act [10]. This deeds registration system is dependent on the preparation of diagrams that detail the dimensions, area and position of approximately nine million land parcels, with the support of figures delineating the shape in a prescribed format [9,15] (*vide* Figure S6 [16] in the Supplementary Material and extract in Figure 6).

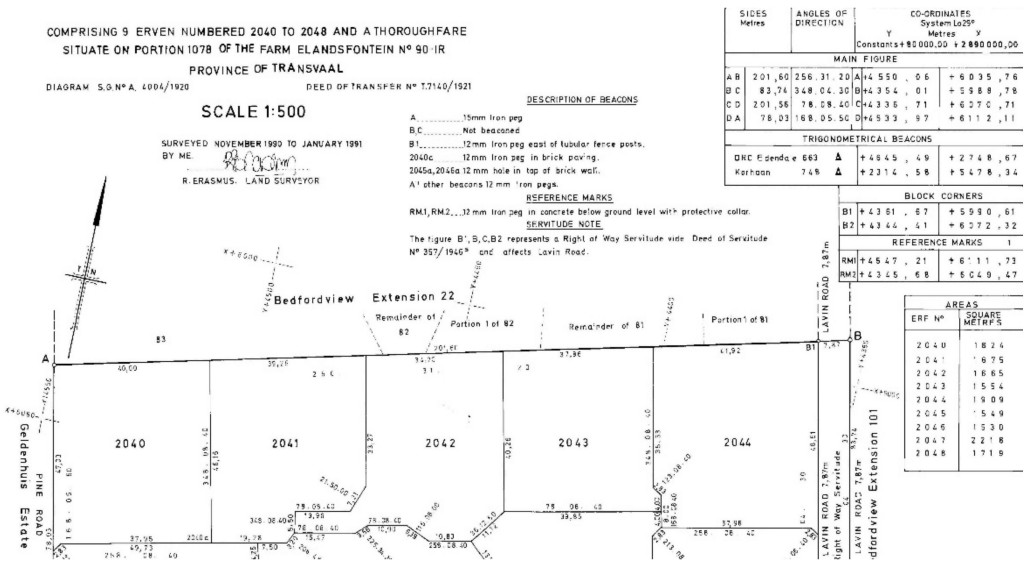

**Figure 6.** An example of a general plan indicating 9 erven in the formal cadastral system in a suburb known as Bedfordview, City of Johannesburg, in the Gauteng Province. Documents such as this are readily obtainable from the Cadastral Spatial Information dataset [16] (Ref: S.G. No. A. 1282/1991).

All diagrams of the South African cadastral system must now be based on points monumented (beaconed) in the ground and coordinated in the National Control Survey System (NCSS) [9,19] that was established and is maintained by the Chief Directorate: National Geospatial Information (NGI) based in Mowbray, Cape Town [17,18]. The NCSS evolved from its origins in about 1860 to its current state, which provides a complete National Reference Framework of base stations. Base stations include Trigonometrical stations (pillar beacons), Town Survey Marks (submerged under an inspection cover, usually at road intersections in urban areas) and, more recently, Continually Operating Reference Stations (CORS) that provide real-time and post-processing reference positioning data derived from GNSS transmissions [20]. This system has also been adopted by many of South Africa's neighbours.

The surveying of land parcels is governed by the Land Survey Act [9]. This Act defines a diagram as "a document containing geometrical, numerical and verbal representations of a piece of land, line, feature or area forming the basis for registration of a real right" [9].

Section 11 of the Act instructs that a land surveyor shall "carry out every survey undertaken by him or her in accordance with this Act, and in a manner that will ensure accurate results and be responsible to the Surveyor-General for the correctness of every survey carried out by him or her" [9]. The Act continues in Section 14 by stating that "no diagram of any piece of land shall be accepted in any deeds registry in connection with any registration therein of that land, unless the diagram has been approved by the Surveyor-General" [9]. Section 16 states that "no diagram shall be approved by the Surveyor-General unless it is prepared under the direction of and signed by a land surveyor" [9].

The system permits rectilinear, curvilinear and ambulatory boundaries between physical beacons. Most of the boundaries delineated in the Cadastral Spatial Information datasets are rectilinear boundaries, consisting of straight lines between the monumented points. (An extract of an area of rectilinear boundaries of the Cadastral Spatial Information database is shown in Figure 7 superimposed over rectified imagery supplied by NGI [12,20].) The line between the monumented points may also be curvilinear—a mathematical curve, a permanent, static topographical feature such as a wall or a fence, or even a dynamic topographical feature such as the middle of a river or the high-water mark of the coast.

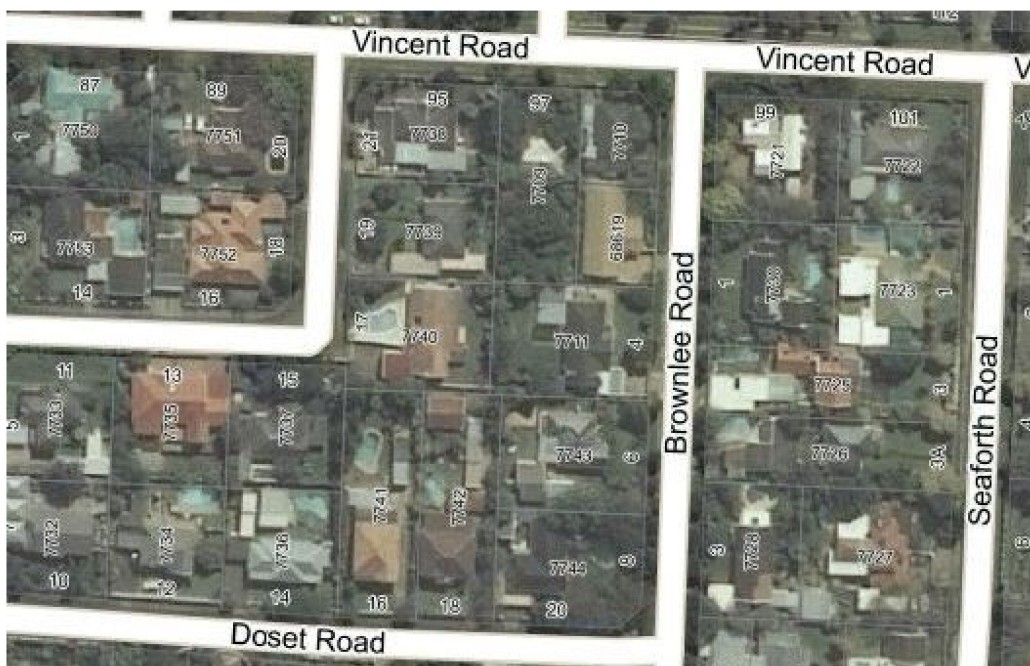

**Figure 7.** 1map Online GIS [21] overlay over rectified imagery of a suburb known as Vincent Heights in a formal cadastre area in East London, Eastern Cape Province, at scale 1:2000 (approximate).

Cadastral Spatial Information has been created from a collection of the delineated boundaries surveyed over 300+ years as documented on the approved diagrams and survey records preserved in the Offices of the Surveyors-General. Each boundary line contained in the Cadastral Spatial Information database is only as accurate as the survey of the original diagram, or any subsequent resurvey of that boundary. Before 1929, standards for the survey of boundary lines were not specifically legislated. Therefore:

- Some early diagrams show boundary lines with no recorded mathematical data— these diagrams only indicate a drawn figure, complemented with the intended area of the land parcel;
- Boundaries may have been shown in relation to topographical features, such as the top of a hill, or following a river or the coast;

- Some of the data defining boundary lines are inaccurate due to poor survey practice and substandard equipment, or were simply paced, ridden, sketched by eye or drawn from memory;
- Many monuments (beacons) defining each end of the boundaries have disappeared completely, resulting in uncertainty of the legal position; and
- Occasionally, land surveyors or the Surveyor-General discover errors in mathematical data or overlaps of diagrams, which must be corrected.

The "general plan" (which is a composite diagram of several land parcels) shown in Figure S6 [16] in the Supplementary Materials indicates the position and extent of each land parcel recorded in relation to the NCSS and adjoining properties to an accuracy of a few centimetres. Once approved, these records are filed in the offices of the Surveyors-General. Each land parcel is then linked to a legal person or entity through the registration of a deed. Section 2 of the Deeds Registries Act [10] gives the Registrar of Deeds the powers and responsibilities to preserve all records of ownership or rights in land, based on the land parcel information held by the Surveyor-General. Further, the Registrar of Deeds oversees the examination and registration of any new deed lawfully submitted to his or her registry for transaction. The deed inextricably links land parcels to people by adding the "who" and the "how" of the rights to a land parcel. A land parcel only becomes a legal object once it has been registered in the Deeds Registry. All such legal objects form the foundation of the South African Land Administration system.

## 5. Analysis in Terms of the Institutional Framework

It cannot be emphasised enough that communities are more likely to preserve, protect and manage their rights when such rights in land are recognised. The current disparate land administration system does not make this a reality for all. The occupants of settlements such as the ones shown in Figures 8 and 9 (Figures S8 and S9 [12] in the Supplementary Materials) remain excluded from the current system. The author has demonstrated in previous research [18] that South Africans with less formal and insecure tenure want their legitimate occupation recorded and recognised. In almost every community that the author has worked for, with and in, the communities have been exemplary in their commitment to the exercise of recordation of their land rights, when the exercise is preceded by community focused and protocol-based communication. People want to participate in the process to ensure an equitable distribution of land rights.

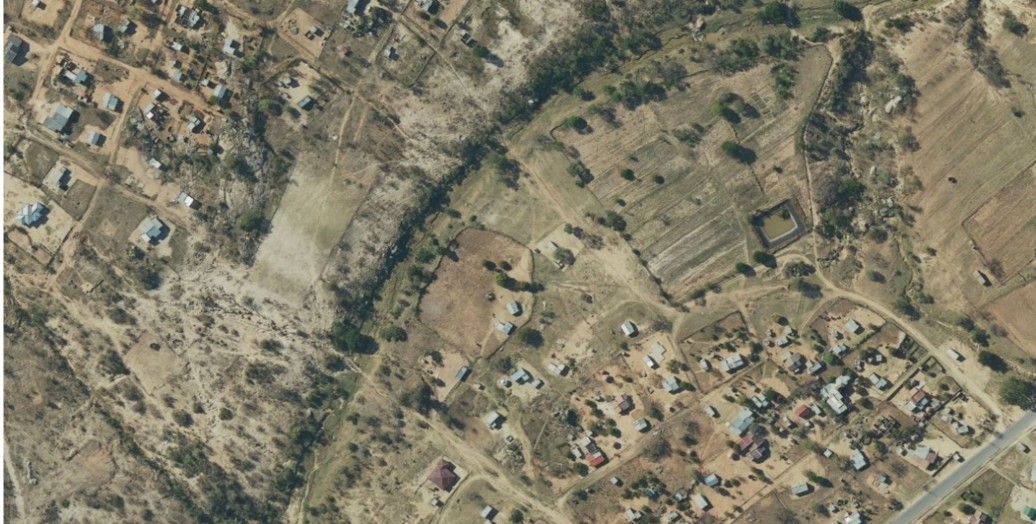

**Figure 8.** Extract from 2018 rectified imagery [12] (Ref: 2531 AD 16), showing an informal settlement in an area known as Luphisi, in Mpumalanga Province, at scale: 1:10,000 (approximate).

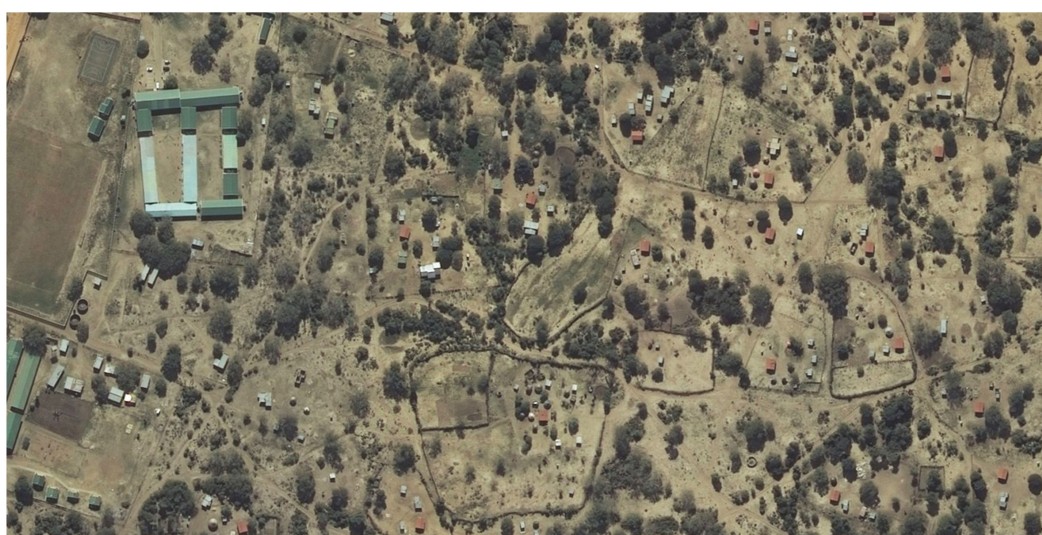

**Figure 9.** Extract from 2019 rectified imagery [12] (Ref: 2732 AB 15), showing part of a rural settlement in an area known as Siphondweni, on communal land in the KwaZulu-Natal Province, at scale: 1:5000 (approximate).

During the 20th century and notwithstanding its bifurcated application, the South African Cadastral System established itself as a high-quality land information system [11], (pp. 225–327). It is currently in decline. Most land administration institutions in South Africa are inadequately resourced; many professional and technical officials trained in spatial information systems could not be retained within the administration. Nevertheless, the cadastral survey adjudication process and the Cadastral Spatial Information databases are still being run remarkably well on outdated systems, by willing but largely inexperienced and ill-equipped officials. Attempts to apply technological innovation, modern systems and updates to operations to improve efficiency, quality and accuracy have encountered many challenges and costly delays. The fit-for-purpose land administration system being proposed will include a prescribed institutional methodology of eight key protocols, some of which can run concurrently. These are set out in Table 1.

**Table 1.** A proposed institutional methodology of eight key protocols.

| No. | Title | Description |
| --- | --- | --- |
| 1 | Initial engagement with the community | Initial engagement with the community will determine the status of the settlement. Is the occupation a traditional community where their land is held in trust by the state or the Ingonyama Trust? Does the community have traditional, civic or social leadership? Does the community have a formal or recognisable identity? This will lead to the determination of position and extent of the settlement and how it is represented on official documents (filed in the Office of the Surveyor-General) of the underlying land parcels as recorded in the existing cadastre. |
| 2 | Determination of current formal land records | Determination must be made of who the land owners of the identified land parcels are by scrutinising official deeds records filed with the Registrar of Deeds. Dispossessed and successors in title of deceased land owners may require legal expropriation as prescribed in the relevant legislation. Where there is no record, the land will probably be unalienated state land (i.e., never registered) as decreed in the colonial annexation of the Southern African territories mentioned earlier. |
| 3 | Determination of current informal land records | Much preparation is necessary to ensure that the relationships, rights, protocols and culture of the community are understood and upheld. Community participation will assist with the determination of any additional rights overlapping formal ownership, whether registered, recorded, social or recognised. These rights could be filed in many places, such as the Magistrate's Office, in offices of the Department of Agriculture, or kept by the Traditional Council. There are also existing rights held by members that may not be recorded in any official archive. (It is always a marvel when, on inquiry, a member of the community produces a well-preserved document that had been issued to a forefather many generations before. Figure 4 is such an example.) |
| 4 | Project planning and funding | It is not only the members of the community that are to be included in the process: relevant government institutions have a vested interest. As part of the preparatory work, those organs of state empowered with developmental functions must determine the availability of funding. This will also have to be presented to the community for agreement together with a detailed, defined project plan. Some funds may be allocated to a communal trust fund to assist the community in supporting the implementation. In certain instances, compensation may need to be considered to acquire land under existing land rights for the provision of services. |

**Table 1.** *Cont.*

| No. | Title | Description |
| --- | --- | --- |
| 5 | Stakeholder engagement | Information sessions must be widely advertised (in writing and by word of mouth) to ensure maximum participation and involvement of the community. Any source of information on community membership should be sought. This may range from records of heads of households held by the leadership, to verbal nomination from participants from the community. It is noted here, that the members present should be requested to consider those not present, including those members not in permanent residence. On state land, the Minister (as the nominal owner) ultimately approves the transfer of land, or the issuing of rights. The Minister must therefore be convinced that all processes have been followed and that the community has been adequately consulted. A generally accepted principle in government circles is that 80% of the community must support or agree to the proposal and such decisions must be recorded in official community resolution documents. |
| 6 | Identification of land occupation | Thereafter, the names of community members can be linked to the position of each homestead and the extent of any land occupation. Using readily available rectified imagery [12] based on the NCSS [19,20], the limits of occupation, use and other rights attached to each homestead can be identified in conjunction with inputs from the community. It is always important to ensure the community's participation in the identification of all boundaries and not to assume that what is visible is the recognised extent of any right. While applying this step, recognition must also be given to non-allocated areas within the settlement and questions should be asked as to who holds the rights to those areas? |
| 7 | Institutional capacity | The institutional framework must include an investigation into the capacity of the administration to process and maintain the increased numbers of land rights and land transactions [5], (p. 7) that will result from the implementation of this inclusive land administration system. |
| 8 | Ongoing maintenance of the system adopted | Lastly, the institutional arrangements must include processes by which people are able to communicate transactions of any land right with the responsible authority. Any legitimate change to the people-to-land relationships must follow a simple process to ensure that the system administrators are able to ensure all land records are current, correct and complete. |

## 6. Analysis in Terms of the Legal Framework

Any fit-for-purpose land administration implementation strategy must include a formal legal framework, which satisfies the political mandate of the ruling party, and which should ideally integrate all land information into a single system to reduce cost and improve access to information. New systems and policies are being considered but will take time to implement. The one key area that is still being debated is whether there should be a differentiation (in terms of the continuum of land rights) between recordation and registration of land rights. This is particularly sensitive in South Africa because of the discriminatory processes imposed on previously disadvantaged groups in the country as described in Section 3. For example, in Figure 9 (Figure S9 [12] in the Supplementary Material), there may well be a diagram and deed for the school site visible on the upper left, but it is highly unlikely that the occupations visible over the rest of the image (some surrounded by hedges) would currently have any form of secure tenure. Similarly, in Figure 10 (Figure S10 [12] in the Supplementary Materials), the visible agricultural units may have some form of recognised right, but there are no formal records of those rights in the offices of either the Surveyor-General or the Registrar of Deeds.

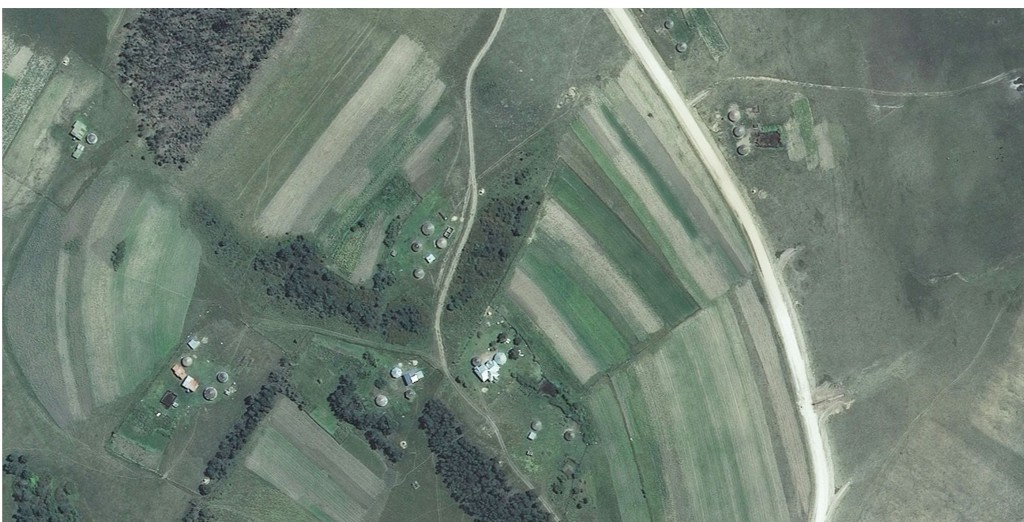

**Figure 10.** Extract from 2015 rectified imagery [12] (Ref: 3130 AA 11), showing small-scale farming in a rural area known as Mahana, on communal land in the Eastern Cape Province, at scale: 1:5000 (approximate).

Current thinking is to separate recordation from registration:

- Recordation would document the reality of land occupation, land use, rights in or to land as it is on the ground. For example, it could include social tenure as is conceptualised in the Social Tenure Domain Model [22]. It would recognise the identity of a person where their right to a specified piece of land is undisputed, recognising that person as a participant with individual, community, communal or informal rights. Recordation (possibly in the form of a Certificate of Land Right to provide a legally secure tenure) may not necessarily proceed to registration as currently prescribed in law.
- Registration (title deed or other forms of legal tenure [10]) would be retained for currently registered land parcels, formal rights, state land, external boundaries of land for identified communities and any applicable legislation that improves rights of insecure tenure (e.g., Upgrading of Land and Tenure Rights Act [23], and Interim Protection of Informal Land Rights Act [24]).

Most important in the South African context is that, while the recordation process will be cheaper and quicker than the requirements of the existing cadastral system, it must not be inferior or less secure. Current legislation will require only minor adjustments to

apply the principles of fit-for-purpose land administration. These minor adjustments are primarily because legislation has not kept pace with innovation, neither has it maintained its efficiency or simplicity. Such minor adjustments will also improve the quality and accuracy of the existing cadastre.

Beyond the legislation controlling the cadastral system, other legislation affecting land rights and land development will need to be assessed. For example, Section 6 (1) (b) of the Land Survey Act requires the Surveyor-General to ensure any diagram or general plan to be approved by him or her is "in accordance with any statutory consent in so far as the layout is concerned" [9]. The requirement of a "statutory consent in so far as the layout is concerned" frequently causes much hardship (time and cost) to developers and prospective beneficiaries of enhancements to land tenure. Scores of pieces of planning legislation remain on the statute books, and are administered by inexperienced and ill-equipped officials, insufficient in number and skills to assess any development of land in terms of the relevant statutes. New legislation, such as the Spatial Planning and Land Use Management Act [25], has been promulgated without the necessary resources to implement the legislation, especially in rural areas, and without repealing old order legislation. This has resulted in frequent duplication of requirements.

Statutory consent applications require extensive research, even if the development is an in-situ upgrade of a fully functional community. Planning and application fees are costly, prescribed circulation through multiple organs of state supplying services is lengthy and often inordinately delaying and the conditions ultimately attached to the statutory consent are often very onerous. Burdensome standards and expectations are frequently imposed on an existing, functional settlement, where all that the members of that community want are their rights recognised and recorded. Much can still be done through the repeal of outdated legislation and amendment and updating of useful legislation to accommodate technological advances and prevent duplication.

## 7. Analysis in Terms of the Spatial Framework

With extraordinary advances in recording methods using GNSS, rectified imagery (such as the extract shown in Figure 11 and Figure S11 [12] in the Supplementary Materials) and GIS, it is suggested that anything produced using a combination of these innovations will be more accurate than many of the existing land records existing in the Offices of the Surveyors-General.

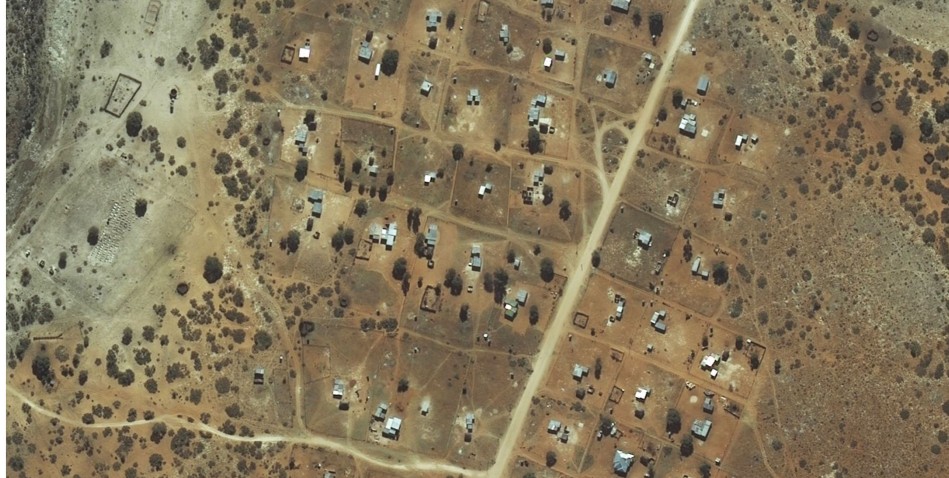

**Figure 11.** Extract from 2019 rectified imagery [12] (Ref: 2723 AB 7), showing an informal settlement known as Molomowapitsana, in the Northern Cape Province, at scale: 1:5000 (approximate).

Strides must be made to replace antiquated systems and equipment with their highly efficient modern successors. NGI, as the South African National Mapping Agency, has extensive aerial photography coverage of the whole country [12]. The denser the population,

the larger the scale of the photography and more frequently updates are undertaken. Much of this aerial photography has been adjusted into rectified imagery at an accuracy better than most of the spatial data recorded in the Offices of the Surveyors-General. As can be seen from the recent rectified imagery of which extracts are recorded in Figures 1, 2 and 8, Figures 9–11 (Figures S1, S2 and S4–S7 [12] in the Supplementary Materials), many of the unregistered land occupations are defined by fences, hedges or walls. Therefore, the country has usable rectified imagery from which all visible boundaries can be determined.

Further, there are also many instances where the original position was unclearly defined, either in terms of description or position. It is suggested that the fit-for-purpose land administration concept can assist in developing standards to guide the identification of positions of many existing boundaries using currently available resources, technology and datasets. The result will be more accurate cadastral records at much lower cost. For example, it is proposed that, where topographical features are used as boundaries, rectified imagery could be used to update those visible boundary positions, especially where they are of a dynamic nature, such as along rivers or the coast. The rectified imagery provided by NGI [12] would satisfy all the prescribed standards of land parcel delineation [9,16,19,20].

## 8. The Way Forward

As with any transformative policy implementation, there are major issues to be considered. Fit-for-purpose land administration recommends the possibility of an incremental approach, where the initial recording tenure rights "using simple and low-cost approaches . . . should be upgraded when need arises" [7], (p. 6), and is therefore built around the flexible recognition of different forms of land tenure.

Much preparatory work is necessary in identifying existing rights on any piece of land, including ownership, unrecorded succession of title, permissions, allocations, occupations and descendants. It is not as onerous as it sounds, as there are generally only two sources of information, firstly the official records held by the current administration (notably the Offices of the Surveyor-General [9,16] and Registrar of Deeds [10]) and, secondly, the institutional knowledge and preserved evidence of the community. It is strongly recommended that a more pervasive Land Rights Enquiry system is developed and implemented within state structures to resolve the plethora of overlapping and conflicting land rights. Already, some of the state-owned enterprises, such as the Electricity Supply Commission (ESKOM) and the South African National Roads Agency Limited (SANRAL), have internal structures engaging with the Surveyors-General and Registrars of Deeds to resolve land issues due to their need to negotiate acquisition of land for their infrastructure that traverses the communal areas.

Another matter to consider is that, during the 1980s, the state issued contracts to land surveyors to document many of the previously un-surveyed dormitory townships that had been laid out some distance away from the formal "white" towns, primarily as accommodation for African labour who worked in the nearby towns. A few hundred land surveyors surveyed hundreds of thousands of land parcels in a very short time and completed the project well ahead of the anticipated schedule. The townships had been laid out, houses built, and yards fenced. In most instances, land surveyors were able to survey the fence corner posts, i.e., boundaries were defined by physical features approximating rectangular sites and blocks as originally demarcated. Figure 12 is an extract of a survey in which the author was involved (Figure S12 [16] in the Supplementary Materials).

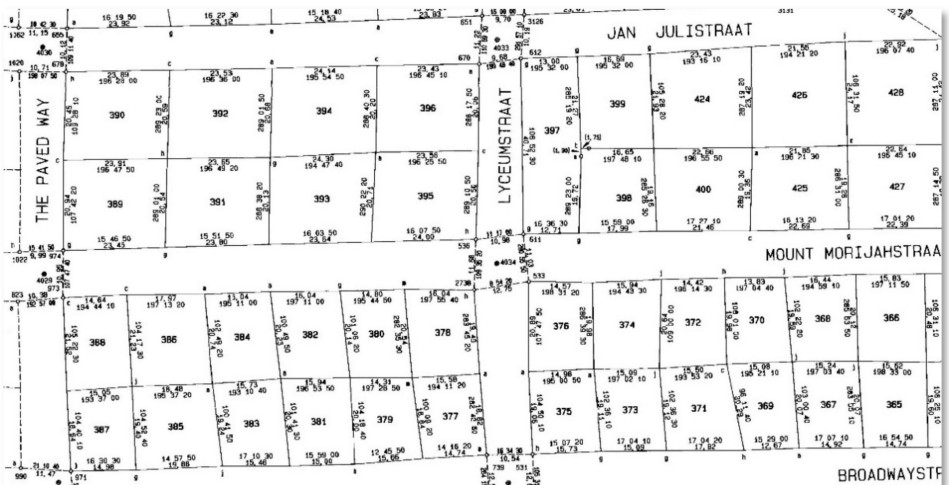

**Figure 12.** Extract from a 1985 general plan of a township with 1092 erven known as Makeleketla, Administrative District of Winburg, Free State Province [16] (Reference: S.G. L. No. 854/1985). The line intersections were determined by fixing the positions of the fence corner posts.

The land surveying team completed the fieldwork of all 1092 erven in a little over two weeks, using the technology available then (single-second theodolites, electronic distance measuring equipment and tape measures). Using more efficient modern technology, such as GNSS equipment, rectified imagery, drones (UAVs) and GIS software, the land surveying profession is confident that the six million land parcels still to be documented can be surveyed to the prescribed accuracies and standards set out in the Land Survey Act [8] and Regulations in an even quicker rate.

Regulation 3, appended to the Land Survey Act [9] standardising field measurements and observations, prescribes that "a land surveyor shall determine the positions of all stations and beacons within the limits of accuracy prescribed in regulation 5 and shall check every part of his or her survey". With regard to determining positions by photogrammetric methods, the standard prescribed in the Regulation is based on out-dated methods. Nevertheless, the facility to use aerial photography is already written into the South African spatial legislation and can be simply updated. The profession is ready and willing to prepare the necessary land parcel documentation, starting with the mark-up of rectified imagery and the creation of any prescribed cadastral plans.

With the quality of the available rectified imagery from NGI [12], many of the land parcels can be delineated straight from the imagery. For example, settlements such as those of Buyelani (Figure 1), Tshiungani (Figure 2), Luphisi (Figure 8) and Molomowapitsana (Figure 11) would require only cursory ground truthing (community participation), as the boundaries of the land parcels are sufficiently recognisable for recordation purposes from the imagery. On the other hand, settlements such as those of Siphondweni (Figure 9) and Mahana (Figure 10) may require more extensive verification on the ground, because hedges and edges of cultivation may not define the full extent of the occupants' rights. Experience of the author has shown that the members of the community will readily provide much of the additional information whenever trust is established.

The buy-in from the community members will also ensure that land administration records are maintained in the future. Already, government has considered "Community Information Centres" to be placed within every municipal area to facilitate ease of access by the communities. Even though it was acknowledged in Section 5 that the Cadastral System is in decline, resources are available to overcome this through state facilitation. Current "Information and Communication Technology" will make it possible for beneficiaries to participate in a web-based fit-for-purpose land administration system.

There are three issues that still require attention.

### 8.1. Institutional

A participatory and transparent method of ascertaining the link of people (both those who reside on the land parcel and those who have rights thereto) to the land parcel must be instituted. A community-based process is essential to ensure maximum participation and reduce the risk of excluding any rightful beneficiary. Any such recordation needs to be accepted into a fully capacitated Land Administration system and any subsequent transaction must be easily updated in that system.

### 8.2. Legal

The state needs to create the legal mechanism of recording the people-to-land relationships, whether it be the current registration system or another form of recording. In addition, the requirements of the plethora of national and provincial planning legislation and development controls, much of which is obstructive to any system of recording of existing land rights, would need to be addressed and overcome.

### 8.3. Spatial

Datasets, innovative technology and human resources are readily available, all of which will facilitate the implementation of the appropriate consultation, delineation and recordation processes. Controlling authorities must still give the go-ahead to utilise them.

## 9. Conclusions

The standard accuracy requirements and the requirement to use registered land surveyors to oversee the delineation of land occupation is not an expensive option. Extensive surveys based on streamlined processes, modern technology, participation by the communities and identification of existing visible features can and will be performed at very little cost and at great speed and precision—"cheap, accurate and fast". The legal requirements of existing cadastral surveying protocol can easily be achieved at very little cost per land parcel.

Government recognises the slow pace at which land registration is implemented in the current system and acknowledges that the cost of acquiring a title deed is too high for most. The government is therefore investigating a new, more inclusive, Integrated Land Administration System that will facilitate the recordation of the large number of new land rights.

In South Africa, there is generally political will to make it happen. Minimal amendments to legislation controlling the cadastral system are required. An additional form of deeds registration (or recordation) is a possibility. Rationalisation of impeding planning legislation is essential. It has been proven that people want their rights documented. Many boundaries are visible on current aerial imagery. Technology exists and is available to provide a high level of accuracy at minimal cost. The land surveying profession is well-established. The resultant land rights would easily be upgradable to formal title. There are therefore many positive aspects already in place and no outstanding issues are insurmountable. A fit-for-purpose land administration system should be implemented in South Africa so that land administration will be "designed to meet the needs of the people and their relationship to land, to support security of tenure for all and to sustainably manage land use and natural resources" [6] (p. 5).

The fit-for-purpose land administration system considered in this paper may assist to formulate recommendations in countries or regions with similar issues. Solutions proposed for South Africa may well find application in many other countries.

**Supplementary Materials:** Rectified imagery is available online on the link http://www.cdngiportal. co.za/cdngiportal/ [12]. Larger images and hence better quality copies of the following rectified imagery extracts are available as Supplementary Materials online at https://www.mdpi.com/article/ 10.3390/land10060602/s1: Figure S1: Buyelani; Figure S2: Tshiungani; Figure S4: Luphisi; Figure S5: Siphondweni; Figure S6: Mahana; Figure S7: Molomowapitsana. Scanned images of diagrams

and general plans approved in the Offices of the Surveyors-General are available online on the link: http://csg.dla.gov.za/ [16]. Images of the complete document from which the following extracts are available as Supplementary Materials online at https://www.mdpi.com/article/10.3390/land10060 602/s1: Figure S3: Bedfordview; Figure S8: Makeleketla.

**Funding:** Funding of the publication costs for this article has kindly been provided by the School of Land Administration Studies, University of Twente, in combination with Kadaster International, The Netherlands.

**Institutional Review Board Statement:** Not applicable.

**Informed Consent Statement:** Not applicable.

**Data Availability Statement:** Aerial photographs are extracted from the rectified imagery dataset of the Chief Directorate: National Geospatial Information, to whom Copyright is acknowledged. The extracts are reproduced under Government Printer's authorisation (Authorisation No. 11841 dated 30 November 2020.

**Acknowledgments:** The author acknowledges much support given by work colleagues and friends in academia in providing a platform to discuss and argue the concepts contained in this paper. In addition, the author acknowledges delegates from member states participating in the Fédération Internationale des Géomètres (International Federation of Surveyors') gatherings over the last seven years who have given the author the original ideas, research material, encouragement and guidance. Some of these are the guest editors of this Journal.

**Conflicts of Interest:** The author declares no conflict of interest. The funders had no role in the design of the study; in the collection, analyses, or interpretation of data; in the writing of the manuscript, or in the decision to publish the results.

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
