# Peer review of "Applying the Fit-for-Purpose Land Administration Concept to South Africa"

_land, doi:10.3390/land10060602_

Round 1

Reviewer 1 Report

Whilst here is merit in pursuing research to evaluate national systems of land administration, tenure and title, this needs to be done in a coherent, rigorous and systematic way, which unfortunately was not the case for the study in question.

The article seeks to apply 'fit for purpose' concepts which is presented as a given from the start but only fully introduced and explained in Section 4.  There was no systematic critique or justification for its use.  Perhaps it would be better to set this up as 'the lens' through which RSAs land administration and title cadastre is evaluated?

On the second page the article abruptly jumps to RSA whereas what the article needed to establish were general principles of land tenure and registration of title from an international perspective before introducing RSA as a case study along with justification of this, perhaps as an example of advanced developing country, against which comparison could be made?

The content on pages 2-7 contain too much descriptive detail which is more appropriate either later on, as evidence to support evaluation or to go in appendices.  Instead the author needs to present a clearly structured and coherent review of literature and theory to then identify the theoretical or analytical framework (Fit for purpose) that they are going to adopt for their study. 

There then needs to be a clear methodology section before embarking on the case study which could begin with the chronology of the evolution of land administration in RSA.  Section 3 is currently inadequate in this respect. Exactly what methods were used to gather what data from where/whom and how was it analysed? Content presented under the 3 frameworks in Section 3 (institutional; legal; spatial) could be consolidated with content in Sections 4.1-4.3 or there needs to be a clearer separation of generic and case study material. 

Consequently the discussions and conclusions were unconvincing and did not flow from a clear, logical and systematic analysis of data to generate original findings.  A lot of the content in discussion section is descriptive rather than higher level evaluation.  The author needs to more clearly demonstrate data, methodological, internal and external validity of their research. 

The article was too long and suffered from poor structure, sequence and balance of content.   The article contained too much detailed and heavily descriptive content and insufficient clarity as to the research methods used, how (empirical?) data was collected and analysed to generate findings.

Other recommendations regarding presentation: 

The 8 key protocols would be better presented/summarised in a table

There were too many uncorroborated  or sloppy statements e.g. 'rough estimates speculate     '; 'some doubts to claims....'  ; 'it has been proposed to support....'; 'but at the end of the day...'; 'was recognised as one of the world's premier cadastral...' 

Reviewer 2 Report

The submitted paper describes a possible concept of FFFPLA to South Africa. The general impression of the reviewer is that this submitted paper fits very good in this journal.

Nevertheless this general impression needs further be underlined by more work on this research paper focusing on the following points:

  1. The overall methodology needs to be improved and should be referenced back in the paper so that the reader is guided along
  2. Overall the synthesis on the findings needs to be improved and so build stronger arguments - e.g Under 3.2, 3.3 and 3.4 key considerations for the institutional, legal and spatial frameworks are described. Then there is an analysis of those 3 frameworks under 4.1, 4.2 and 4.3 and then in the Discussion again under 5.1, 5.2 and 5.3 is organised around the 3 frameworks - which is generally really good. But the reviewer would suggest to the author to either improve the synthesis by explaining the reader (preferable in the methodology section) how the key consideration flow into the analysis and then flow into the discussion. This could be done by the methodology or by stating it more clear in the text.

Some major/minor comments and observations:

  • Subsection 3.2.2 needs special attention during the implementation of FFPLA in South Africa and issues such as land grabbing (as described in a lot of UN reports) at ‘unoccupied’ land needs to be addressed and considered beforehand.
  • In subsection 3.2.3 reference could be made to the ‘source’ in LADM/STDM and mentioned which kind of source document would be important to be accepted
  • In subsection 3.2.6 the question of overlapping rights and its legal implications could be considered

    Further it is mentioned the Resident Magistrate under the formal “Per-mission to Occupy” (PTO) system the reviewer is wondering if it could/should be considered to integrate data stored there and in general if data collected from other institutions and or NGOs could and should be integrated - that would have legal but also institutional and spatial implications. Probably the author could elaborate on that or consider the pros and cons of such an approach. But also e.g. how to integrate data form Electricity 604Supply Commission(ESKOM) and the South African National Roads Agency Limited605(SANRAL), and would it be worthwhile to include those.
  • The reviewer noticed a little too less focus on capacity especially when it is about the institutional framework (3.2) it is slightly addressed in 4.1 but probably could be further elaborated in 3.2 also.
  • Further though it was not mentioned in the article but the reviewer was wondering if it would be worthwhile to also mention about the costs of titles and especially since there are a lot of customary titles how those costs would/could be covered.
  • The cost aspect is especially also relevant when reading that ‘In South Africa, the physical position of a boundary on the ground (whether a beacon or a topographical feature) takes precedence over the mathematical and documented position, unless it can be proven that the physical position has been moved unlawfully’. This is a very interesting aspect which bears a lot of risks and is not completely in line with an FFPLA approach.
  • On P.15 line 555 it says - It id not registered in the Deeds Registry. Something missing there?

Reviewer 3 Report

I have found this paper very interesting and relevant in understanding land administration  in South Africa. The historical background is well captured and very informative.

The methodology is well written as well as the other remaining sections.

In general, I am happy with this work and dont have much to add but to thank you for the efforts put in writing it.

Reviewer 4 Report

The problems introduced in the manuscript are typical not only in South Africa, but also in other countries of the African continent, and even the Mexican or the Hungarian land register systems struggle with some of the problems outlined here. Because of this, the topic may be of wide international interest.

However, it is a problem that the author sometimes digs too deeply into the topic (by describing technical information and parameters, too many legal paragraphs), which makes the manuscript longer than usual and more difficult to understand. Certain topics are discussed repeatedly, even if the point of view is somewhat different. The manuscript already contains all the relevant information for a good article, but the structure of the manuscript needs to be reconsidered and the delivery of the messages to the readers is not yet sufficiently mature. On the one hand, the text does not have the right logical arc, on the other hand, it is not compact, the author tries to say too much instead of focusing on the relevant messages. This is why I can only support the publication of the article after a thorough restructuring.

The abstract is perfectly fine, as it adequately summarizes the content and results of the manuscript.

At the beginning of the introduction, the author introduces the general challenge and arouses the reader’s interest in the article. However, this section already goes too deep in presenting the specific problems while necessary elements of an introduction are missing, for example, well-defined research questions, an explanation of why South Africa is a good case study to answer them, and finally the introduction to the structure in which the author will discuss the topic.

Section 2 presents the historical development of the South African cadaster system. Between lines 137-150 and 167-175, there is technical information that does not need to be explained here, it does not add anything to the understanding of the topic discussed later. If possible, I suggest that in this chapter the author only presents the roots of today's problems, with a summary at the end of the section (I recommend integrating sub-section 3.1 here).

In the first sub-section of Chapter 3, the concept of fit-for-purpose land administration should be presented (former sub-section 4.1). I suggest a complete rethink and rewrite of the sub-sections between 3.2 to 4.3 (except 4.1). My reason behind this recommendation, that the ‘Institutional framework’, ‘Legal framework’ and ‘Spatial framework’ sections discuss problems which already discussed, albeit here in more detail, but often repeating each other, so their function in this form is not clear. One can only guess that in Section 4 the author discusses these topics from the perspective of fit-for-purpose land administration, but it is still repetitive in nature. My suggestion is to combine these quasi-double sections into three sub-sections where the first half of each presents the challenges/obstacles of applying the new concept and the second half discusses the possible solutions.

I propose a case study for Section 4, as the author cites many specific examples in the manuscript (with photographs, which makes the problems well illustrated). Instead of scattered pieces of information, it would be useful to illustrate the process of introducing fit-for-purpose land administration through an example of a community or area.

In Section 5, I suggest to the author to focus on the most important 3-5 messages that he wants to convey to his readers and which may be of international interest. It would be useful here to compare the results with the results of similar research from other countries or continents to help formulate a general message to the wider audience.

Overall, the article provides an interesting presentation of the South African land registry system and the problems that arise today. The author’s high level of professionalism and commitment to the topic is evident from the manuscript. At the same time, a scientific paper, even a case study, should not only aim to present local problems and solutions, but also to formulate general recommendations that can be applied elsewhere.

Reviewer 5 Report

Really like the insight into the land administration situation in South Africa with all the corresponding complexities - great to see this described in one article. The methodology need to be reshaped around the eye of the FFPLA three frameworks (spatial, legal and institutional) to structure your analysis and to evaluate what problems FFPLA will solve and identify what problems are the most difficult to resolve. Shorten the article though the use of appendices to describe the history of land administration in South Africa - it is a great narrative. I encourage you to revise the paper as it will be  excellent.

Round 2

Reviewer 1 Report

The author is to be commended for taking on board previous reviewers' comments and making some positive and necessary changes to the article.  The article is better but there are some further improvements that are needed as follows:

Abstract needs re-writing once the final article has been revised; does not start off well with a closed y/n question; surely it should be 'How well does the.....' or 'What potential does....'?

In terms of the main body of the article:

Section 1 is improved but needs to clearly establish the lens/framework that is created by fit for purpose concept that is employed in the study (e.g. institutional; legal; spatial)

Section 2: I was pleased to see this section presented however it still needs further refinement.  Please lose the first list and turn into prose; you need to set our the purpose of the concepts and how the components are deployed as part of the analytical framework to be applied to RSA case study; you also need to make a compelling case as to why study of RSA using fit for purpose is a useful contribution (I believe that it is) - perhaps along the lines of potential to make a real difference compared to other administrations; line 122/123 states 'It has been proposed' by whom? You? Whilst the final paragraph in section 2 signposts what follows it does not clearly set out the methodological approach adopted. 

Section 3 is generally ok apart from occasional missing in text ref e.g. line 176  16 million ha

Section 4: need to get rid of first bullet point list and present either in full prose or a table; in text refs missing e.g. in whose opinion has PTO system floundered? what is the evidence for this?  What follows is very descriptive and I still think that some of the heavier description could be moved to an appendix

Section 5: mention is made of 'solution' but the intro and methodology was not set up in this way - if you believe that Fit for Purpose is a solution then this should be stated from the outset; also to to recognise that the suggested improvements to recoding of land tenure and ownership have not been tested but are proposals therefore somewhat speculative; if there is evidence of any of this having been tested/deployed in practice then need to refer to reported and published evidence;

Table 1 is a helpful addition/device - could be spaced better.  

Section 8 is better framed and sequenced; there is a helpful return to the 3 domains used as the analytical framework - are these more recommendations/proposal that issues requiring further attention as the latter has already been covered in sections 5, 6 and 7? You could consider combining sections 8 and 9 into 'the way forward/recommendations and conclusions?

Can the number of references in the article be increased?   

Reviewer 4 Report

I accept the author's response to my review.

The re-structuring of the manuscript is well done, this new structure is more logically thoughtful, leading the reader in a linear fashion from problem statement to practical solutions. This makes it much more readable and understandable than the previous version.

I recommend the publication of the manuscript in its current form. 

Reviewer 5 Report

The restructuring of the article and associated methodology has significantly improved the article - well done. This detailed investigation into the potential use of FFPLA in South Africa provides your politicians with an excellent opportunity to accelerate security pf tenure and associated social equity. There is only one revision I would propose and that is to provide a statement in chapter 8 'The Way Forward' on how the land rights data can be maintained efficiently in the future.

Author Response

This manuscript is a resubmission of an earlier submission. The following is a list of the peer review reports and author responses from that submission.